environmental chemistry/geochemistry/
environmental science

ion-adsorbed rare earth ore, pore structure,
ionic strength, ion exchange, seepage

**Authors for correspondence:**
Xiaojun Wang
e-mail: xiaojun7903@126.com
Lingbo Zhou
e-mail: lingbo_318@126.com

This article has been edited by the Royal Society of Chemistry, including the commissioning, peer review process and editorial aspects up to the point of acceptance.

# Dynamic pore structure evolution of the ion adsorbed rare earth ore during the ion exchange process

Lingbo Zhou[1,2], Xiaojun Wang[1,2], Yulong Zhuo[1,2], Kaijian Hu[1,2], Wen Zhong[1,2] and Guangli Huang[1,2]

[1]Jiangxi Key Laboratory of Mining Engineering, and [2]School of Resources and Environment Engineering, Jiangxi University of Science and Technology, Jiangxi 341000, People's Republic of China

LZ, 0000-0002-4683-492X; XW, 0000-0003-4664-2386

During the leaching process of the ion-adsorbed rare earth (RE) ore, the pore structure evolution of the ore body plays a vital role in the seepage of the leaching solution. To investigate the evolution of the pore structure during the leaching process, experiments have been carried out with remodelled RE ore samples based on the physical characteristics of *in situ* ores. The seepage rate difference between deionized water leaching solution and 2% $NH_4Cl$ leaching solution during the active leaching period was analysed. The porosity and the dynamic pore size evolution of pore structures in the ore body are discussed. Results indicate that along with ion exchange between the RE ore and the leaching solution, the porosity of the sample remains constant and the pore structure shows a decreasing trend in the first part and an increasing trend in the second part. Specifically, during the ion exchange process, the number of minimal pores (0–5 µm), small pores (5–10 µm) and medium pores (10–25 µm) increases significantly and the number of medium–large pores (25–60 µm), large pores (60–120 µm) and mega pores (greater than 120 µm) decreases dramatically. Along with the completion of the ion exchange process, the evolution of porous structure shows an opposite trend. The mechanism study reveals that the evolution of pore structure is induced by the difference of ionic strength in the leaching solution during the ion exchange process, where the RE ore microparticles will be absorbed or desorbed on to the solid phase.

## 1. Introduction

Ion-adsorbed rare earth (RE) ore, also known as weathered crust infiltration RE ore, is a unique type of RE ore found in China. It is mainly absorbed on clay type ore (i.e. kaolin clay, illite) in the

form of hydrated cations and can release RE cations via the chemical and biological process [1–3]. The resource is characterized by heavy and RE elements with distinctive advantages, such as complete distribution, high added-value, low radioactivity ratio and high value of comprehensive use. RE resources play an extremely important role in the modern high-tech industry, and they have received extensive attention [4,5]. Significant efforts have focused on the development of this valuable mineral resource, such as metallogenic mechanisms, mining technologies, separation and purification, and applications of RE elements in various fields, which provided basic technological support to efficient developments and green extraction of RE elements [6–9]. The *in situ* leaching is the most commonly used mining method of extracting RE elements, during which RE cations absorbed on the ore body are firstly replaced by more active cations in the leaching solution and then extracted from collected mother solutions [10–12].

However, due to the continuous injection of the leaching liquor, the RE ore body is fragile under hydrolysis conditions, and its stability is extremely poor in the presence of a hydrous effluent [13]. In addition, the pore structure inside the RE ore is irreversibly changed by the strong chemical replacement reaction between RE cations and the cations in the leaching liquor. Hence, the chemical and physical characteristics of the ore body are altered, resulting in landslide accidents, casualties and economic losses [14,15]. Previous studies have shown that, during the *in situ* leaching, water is not involved in the chemical replacement reaction; and the mineral composition and content of RE particles hardly change, while the particles are relatively distorted under the influence of fluid [16,17]. With the increase of the leaching solution volume, water seepage has a certain effect on the particle retention, and the particles migrate deeper in the porous media [18]. The most obvious migration of soil particles is present in the lower part of the porous media [19]. The influencing factors of particle migration in porous media include pore structure and seepage velocity, and the larger the seepage velocity, the more obvious the effect of pore structure [20]. During the *in situ* leaching process, the RE ions are recovered by using the ion exchange reaction between the leaching solution and the ore body. The leaching rate of the RE ions depends on the intensity of ion exchange and the seepage of the leaching solution in the ore body [21]. Strong ion exchange will cause the deposition and release of fine RE particles, which will further affect the seepage of the leaching solution in the ore body [22]. Some studies have shown that, after leaching, chemical replacement reaction changes the composition, size and gradation of particles in RE ore, and then changes the strength and other mechanical properties [23]. For instance, the cohesion of RE ore is decreased; the internal friction angle is increased and gradation is changed. On the one hand, the cohesion of RE ores tends to decrease, while the cohesion slightly increases after leaching; on the other hand, the leaching solution can further promote the weathering of RE ores, increase the clay content and reduce the particle size, and leaching can cause soil erosion and vertical migration of particles, easily resulting in landslide [24,25].

Aiming to investigate the seepage and migration laws of leaching solution in the leaching process, many studies, focusing on the influence of water seepage on the fine particles inside the RE ore body, have been reported. However, *in situ* leaching involves two important processes: ion exchange and solution seepage, in which solution seepage is a physical percolation process, and ion exchange is a chemical replacement process. As the ion exchange process involves microscopic ion replacement, however, at present, no study has been done on the influence of the diffusion and the electric field during the leaching process on the porous structure of the ore body. In addition, the change of the pore structure of the RE ore body will affect the seepage effect of leaching solution, resulting in the change of the leaching rate of RE elements. On the other hand, the stability of RE ore body under the influence of solution seepage is very poor, which will affect the stability of the hillside of the RE mine. As the ion exchange process continues, the fine particles in the ore body migrate, which cause the transformation of the secondary pore structure, affecting the osmotic transport characteristics of the leaching solution and the stability of the RE ore body. This may cause safety accidents such as mine landslides. In this paper, we simulated the leaching process with deionized (DI) water and 2% NH$_4$Cl solution as leaching solutions. By comparing the internal pore structure evolution of the RE ore in these two leaching processes, the influence of the ion exchange process on the porous structure of the ore body is revealed and this result will provide scientific methodologies and theoretical supports to the *in situ* leaching of the ion absorbed RE ore.

## 2. Material and methods

### 2.1. Mechanism discussion

The extraction of RE elements from the ore body is an ion exchange process. With the injection of the leaching solution, RE cations absorbed on the ore body will be replaced by more active cations in the

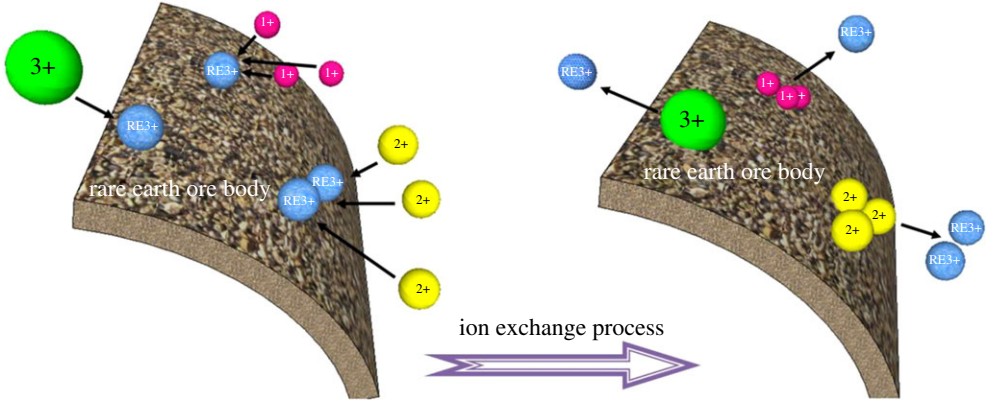

**Figure 1.** The schematic of the ion exchange process.

**Table 1.** Physical parameters of the RE sample.

| parameter type | ore grade (%) | density (g cm$^{-3}$) | moisture content (%) | specific density |
|---|---|---|---|---|
| value | 0.0673 | 1.75 | 13 | 2.675 |

solution seepage process [26–28]. The leaching process involves a variety of chemical reactions. However, most of them are not included in the scope of the research as they are too weak to influence the RE leaching process. The main chemical reaction is shown in equation (2.1) [29,30]. The exchange between the RE cations and more active cations in the leaching solution follows the law of equivalent exchange, in which the equal amount of charges is exchanged (figure 1). This process starts with the diffraction of cations in the leaching solution through the diffusion layer on to the ore particle surface. RE cations will then be replaced and eventually be extracted from the mother solution.

$$[Al_4(Si_4O_{10})(OH)_8]_m \cdot xRE^{3+}_{(s)} + \frac{3x}{y(SE^{y+})_{(aq)}} \rightleftharpoons [Al_4(Si_4O_{10})(OH)_8]_m \cdot \frac{3x}{y(SE^{y+})_{(s)}} + xRE^{3+}_{(aq)}. \qquad (2.1)$$

In this equation, $RE^{3+}$ is the RE cations, $SE^{n+}$ is cations with relative active chemical properties (i.e. $NH_4^+$, $Na^+$, $Mg^{2+}$, $Al^{3+}$).

## 2.2. Sample preparation

Large raw ore samples with a size of $1 \times 1 \times 0.5$ m were obtained from a RE ore mine. The sample collection process started with wrapping the four sides of a sample with bamboo sheets and steel wires. Then, the bottom side was carefully trenched. Finally, wood sheets were inserted into the bottom side and the sample was extracted from the mine. Several samples were collected with the same method and transported to the laboratory with careful protection [31]. The ore grade, density, moisture content and specific density were all measured from raw samples and are listed in table 1. However, these raw samples directly obtained from the mine were not suitable for sample preparation. The testing specimens used in this study were remodelled according to the physical parameters of raw samples. An equal amount of loose RE ore sample, collected from the same mine site, was added to a mould layer by layer. A total of 109 g of RE ore was divided into three portions and filled into a mode separately. In each layer, the ore sample was firstly punched with a hammer twice from a maximum punching distance (figure 2) and then the sample surface was roughed. In order to meet the size requirement of the following leaching experiments and nuclear magnetic resonance (NMR) tests, the diameter-to-height ratio of ore samples was 40 mm : 60 mm. A batch of 30 remodelled samples was fabricated. The initial porosity of these samples was measured by an NM-60-type microstructure analyser based on the NMR technology. A total of 24 samples with a similar porosity were selected for the following experiments. Sample preparation instruments and remodelled samples are shown in figure 2.

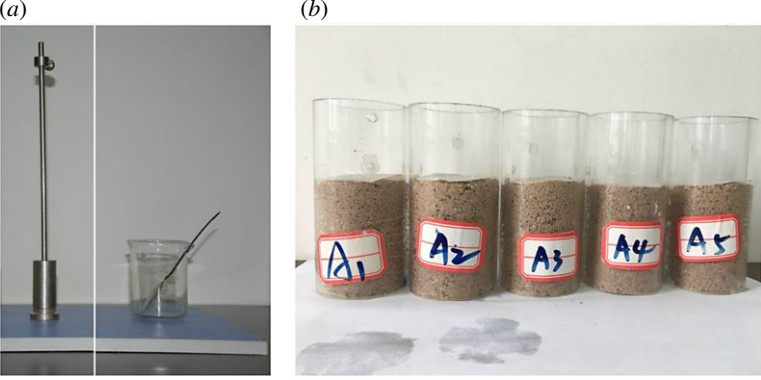

**Figure 2.** (*a*) Sample preparation instruments. (*b*) Remodelled samples.

## 2.3. Experimental procedure

### 2.3.1. Leaching experiment

As shown in figure 3, 10 selected remodelled samples were equally divided into two groups for leaching tests. The experiments were carried out at the same room temperature with the same leaching solution to rule out the influence of temperature and the pH and the concentration of leaching solutions. The pH values of DI water and 2% $NH_4Cl$ solution used in this experiment were 6.92 and 6.10, respectively. In this study, leaching solutions were DI water and 2% $NH_4Cl$ solution. Before the experiment, all testing samples were saturated. After saturation, the volume of the collected solution was roughly equal to that of the injected solution. This process was necessary for the calculation of the data of leaching tests. The DI water was selected to saturate RE ore samples with multiple injecting and collecting cycles. In every cycle, 30.00 ml of DI water was injected, and the volume of the collected solution was measured. If the amount of the collected solution was maintained at around 29.9 ml, RE ore samples were saturated. The leaching experiments were then carried out in cycles by adding 30.00 ml of DI water and 30.00 ml of 2% $NH_4Cl$ solution into two groups of samples in every cycle. Here, adding an equal amount of the leaching solution was to ensure the same amount of ions that are involved in the reaction. Ten samples were tested at the same time and flow control valves in every testing apparatus were set at the same rate. In every cycle, about 29.90 ml of the mother solution was collected and the collecting time was recorded. The concentration of the RE cations was measured via ethylenediaminetetraacetic acid (EDTA) titration, and the pH of the mother solution was also analysed. The endpoint of the leaching test was terminated when the concentration of the collected mother solution was below 0.05 mg $ml^{-1}$.

The remaining 14 samples were also evenly divided into two groups (seven samples per group) for another leaching test. It was consistent with the leaching test mentioned above, one group was leached by DI water, and the other group was leached by the 2% $NH_4Cl$ solution. However, in this test, samples in the same group were leached with different leaching cycles ranging from one cycle to seven cycles. The pore structure of every RE ore sample was measured by an NM-60-type microstructure analyser. Then, these samples were properly stored and dried for later scanning electron microscopy and energy spectrum analysis.

### 2.3.2. Rare earth cation concentration measurement

Concentrations of RE cations in collected mother solutions were determined by titration with EDTA. Two EDTA standard solutions with concentrations of 0.08555 and 0.3393 mg $ml^{-1}$ were prepared. The test started with sampling $E$ ml of a mother solution into an Erlenmeyer flask. Then, 10 ml of buffer solution, 1 ml of sulfosalicylic acid (1 wt%), 1 ml of ascorbic acid (5 wt%), 1 ml of acetylacetone (5 wt%) and two drops of xylenol orange were added in the Erlenmeyer flask in order and the Erlenmeyer flask was shaken well. A standard EDTA solution was added into the Erlenmeyer flask drop by drop via an acid burette until the solution turned a bright yellow colour. The EDTA level in the acid burette was started at $V_0$ ml and ended at $V_1$ ml. Concentrations of RE cations in mother solutions were calculated according to the equation given below

$$C_{RE} = \frac{C_{EDTA} \cdot (V_1 - V_0)}{V_2},$$

(2.2)

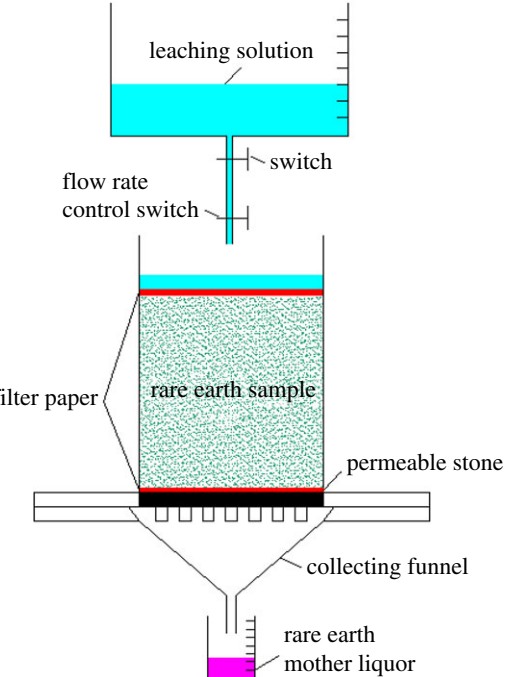

**Figure 3.** Schematic diagram of the simulated leaching process.

where $C_{RE}$ is the concentration of the mother solution, mg ml$^{-1}$; $C_{EDTA}$ is the concentration of the EDTA standard solution, mg ml$^{-1}$; $V_0$ and $V_1$ are the initial and final volume readings on the acid burette, respectively; $V_2$ is the volume of the sampled mother solution, usually 5.00 ml.

In this study, the RE cation concentration, in the mother solution collected in every cycle, was measured. When the concentration of the RE cations dropped below 0.05 mg ml$^{-1}$, the leaching test was stopped. As the concentration of every mother solution in the DI water leaching test was 0 mg ml$^{-1}$, for a better comparison, the number total leaching cycle of DI water leaching test was kept the same as that of the NH$_4$Cl leaching test.

### 2.3.3. Porous structure test of rare earth ore

The NMR technology is a non-destructive detection technology that has been widely used in chemical engineering and other fields. It is through the detection technology of H element, H element is stimulated by radio frequency, energy changes, the time needed to restore to the initial state can reflect the environmental structure of H element. Therefore, the part of the pores occupied by the pore water can be tested by the NMR technology and the nuclear magnetic signal only acts on the H element [32–34]. The pore structure of every RE ore sample of this experiment was measured by the NM-60-type microstructure analyser (Suzhou Niumag Analytical Instrument Corporation, Suzhou, China). The temperature of the permanent magnet was stabilized at $32 \pm 0.1°C$ before the test to ensure data accuracy. After every leaching cycle, the remodelled RE ore samples were placed horizontally on the bracket of the microstructure analyser (figure 4) and the porosity data, porous distribution data and inversion images of each sample were collected.

### 2.3.4. Microstructure unit test of rare earth ore

The microscopic crystal morphologies of RE ore samples were observed by an MLA650F field emission electron microscope and energy spectrometer. The photographs of the test instrument and samples are shown in figure 5. The samples were fixed with epoxy resin adhesive to maintain their structures after leaching. The injected epoxy resin adhesive did not fully cover the entire sample, leaving desired observation sites for inspection. The electron microscopy and energy spectrum analyses were conducted on the location of a sample, where the abnormal signal was observed in its inversion image. For comparative analysis, scanning electron images were also taken at the sample part of a sample leached by DI water.

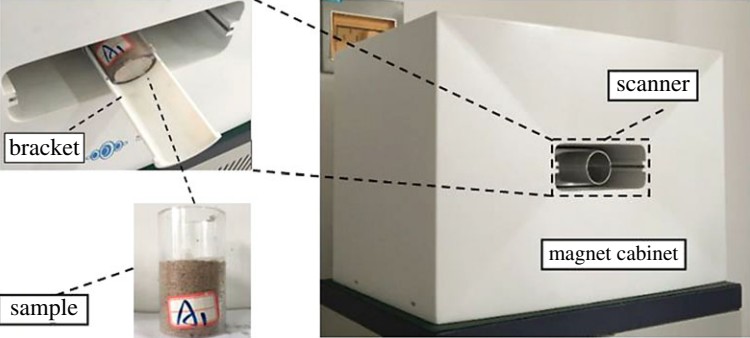

**Figure 4.** The NMR test instrument.

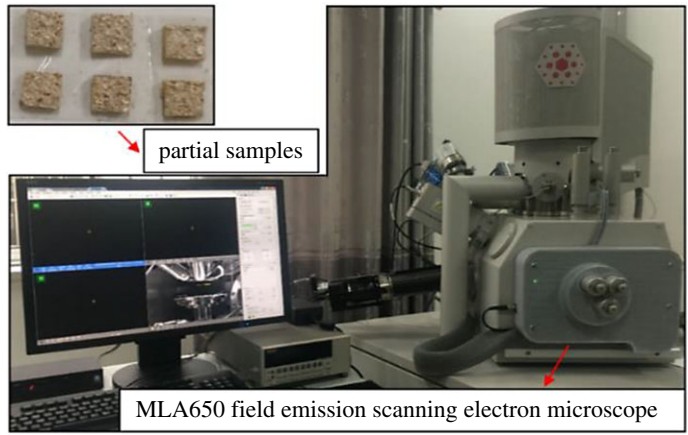

**Figure 5.** The MLA650 field emission scanning electron microscope and partial samples.

# 3. Results

## 3.1. Analysis effective leaching time

The total leaching cycles and leaching time in the 2% $NH_4Cl$ test were determined by the final cycle, where the concentration of RE cations in the mother solution dropped below $0.05\,mg\,ml^{-1}$. As the concentration of the mother in the DI water leaching test remained constant, the cycle number of this test was kept the same as that of the 2% $NH_4Cl$ test for better comparison. The data of leaching tests are listed in table 2.

The 2% $NH_4Cl$ leaching test consists of seven leaching cycles and lasts 14.2 h. In the last cycle, the concentration of RE ions in the mother solution was below $0.05\,mg\,ml^{-1}$, and the leaching test is terminated. The total recovery of RE is 91.757%. The concentration profile of the RE cations in every leaching solutions and mass of the RE elements at different leaching time are all plotted in figure 6. As the variation between the data of different samples in the same leaching test is very small, the data analysis is focused on two samples with the closest porosity selected from DI water leaching test (sample 1) and $NH_4Cl$ leaching test (sample 2). As shown in figure 6, the first two cycles are the saturating period. No RE cation was detected in the leaching solution. This result indicates that the DI water leaching solution has no interaction with rare earth cations. After two leaching cycles (3.1 h), 2% $NH_4Cl$ solution is used as the leaching solution of sample 2 to initiate the ion exchange process. After the third leaching cycle (5.1 h), the concentration of the RE cations starts increasing and reaches a maximum after the fourth cycle (6.9 h). After the fifth cycle, the cation concentration in the leaching solution drops dramatically and reaches a level close to the minimum industrial grate (0.1%) [35,36]. As leaching of the RE cations lags the ion exchange process, according to the concentration profile in figure 6, the ion exchange process mainly takes place in the third cycle and the fourth cycle, where the third cycle is the main reaction period and the fourth cycle is the residual reaction period. Here, the period of the ion exchange process is defined as the effective leaching time. In the 2% $NH_4Cl$ solution leaching process, the effective leaching time is determined from 3.1 to 6.9 h.

**Table 2.** Data and experimental parameters of the DI water leaching test and the 2% NH₄Cl leaching test.

| types of leaching solutions | DI water (saturating period) | | DI water (leading period) | | | | | DI water (saturating period) | | 2% NH₄Cl solution (leading period) | | | | |
|---|---|---|---|---|---|---|---|---|---|---|---|---|---|---|
| no. leaching cycles | 1 | 2 | 3 | 4 | 5 | 6 | 7 | 1 | 2 | 3 | 4 | 5 | 6 | 7 |
| volume of injected solutions (ml) | 30.00 | 30.00 | 30.00 | 30.00 | 30.00 | 30.00 | 30.00 | 30.00 | 30.00 | 30.00 | 30.00 | 30.00 | 30.00 | 30.00 |
| pH value of leaching solutions | 6.92 | 6.87 | 7.10 | 6.86 | 7.06 | 6.96 | 6.93 | 7.02 | 6.96 | 6.10 | 6.08 | 6.05 | 6.06 | 6.13 |
| volume of collected solution (ml) | 15.50 | 29.90 | 29.80 | 29.70 | 29.60 | 29.80 | 29.90 | 15.50 | 29.80 | 29.70 | 29.60 | 29.90 | 29.70 | 29.90 |
| pH value of collected solution | 6.80 | 6.85 | 6.76 | 6.69 | 6.84 | 6.93 | 6.58 | 6.38 | 6.46 | 6.63 | 6.7 | 6.62 | 6.65 | 6.75 |
| leaching time (h) | 1.0 | 1.7 | 1.7 | 2.0 | 2.1 | 2.1 | 2.3 | 1.3 | 1.8 | 2.0 | 1.8 | 2.7 | 2.1 | 2.5 |
| weight of the collected RE element (mg) | 0 | 0 | 0 | 0 | 0 | 0 | 0 | 0 | 0 | 4.83 | 56.04 | 4.30 | 1.37 | 0.77 |
| total testing time (h) | 1.0 | 2.7 | 4.4 | 6.4 | 8.5 | 10.6 | 12.9 | 1.3 | 3.1 | 5.1 | 6.9 | 9.6 | 11.7 | 14.2 |

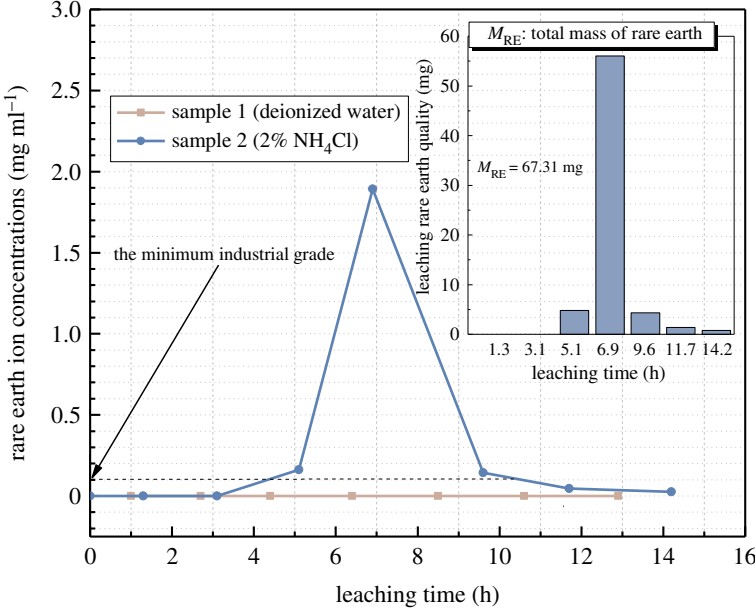

**Figure 6.** The concentration of RE cations in collected mother solutions at different leaching time. Inset chart: the mass of the leached RE element at different leaching time.

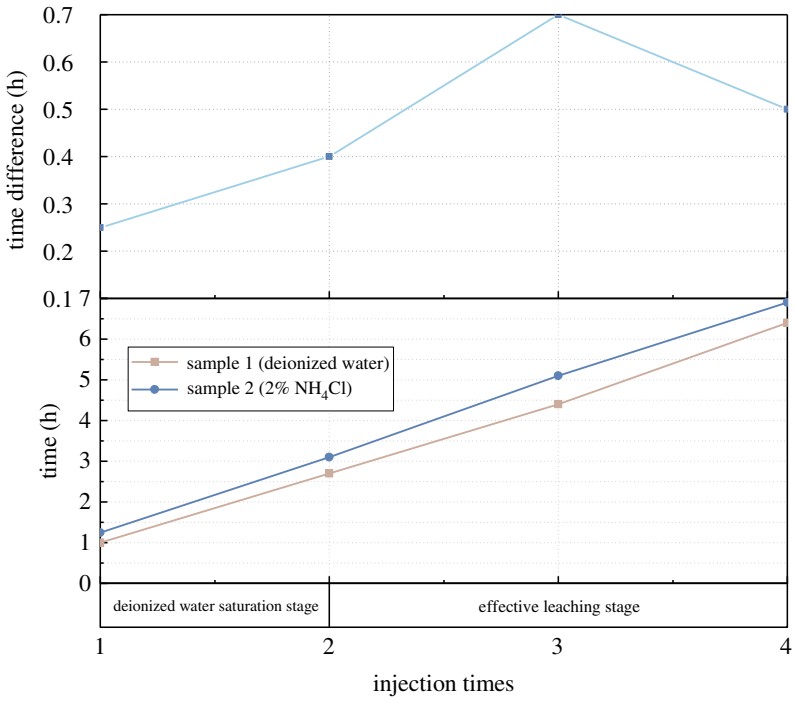

**Figure 7.** Leaching time difference between DI water leaching test and 2% NH$_4$Cl leaching test.

## 3.2. Analysis of leaching time difference in leaching tests

Even with similar starting porosity and injection of the same amount of the leaching solution (30.00 ml), the leaching cycle time of 2% NH$_4$Cl is much longer than that of the DI water. The leaching time difference and leaching time profiles of these two tests are plotted in figure 7. In first two cycles, sample 1 and sample 2 are all in the saturation stage and the leaching time difference is small (less than 25 min), which is mainly due to the minor porosity difference between two samples. After the third cycle, the DI water leaching solution of sample 2 is replaced by 2% NH$_4$Cl solution, and sample 1 is kept using DI water as the leaching solution. The leaching time of sample 2 increases dramatically

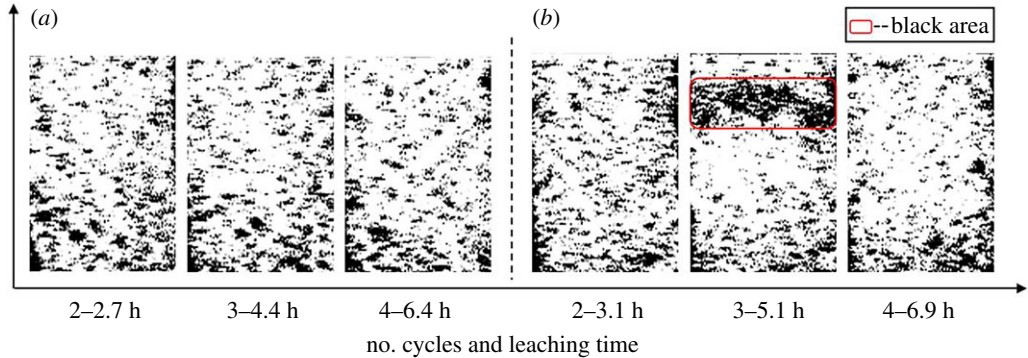

**Figure 8.** Inversion diagrams of pore structures during the leaching process: (*a*) DI water leaching and (*b*) 2% NH$_4$Cl leaching.

compared to that of sample 1. The time difference between these two samples reaches over 40 min. As the third cycle is the main ion exchange period, the difference in the leaching time indicates the ion exchange process reduces the solution seepage rate. In the fourth cycle, the leaching time of sample 1 and sample 2 shows an opposite trend. The leaching time difference between these two samples in this leaching cycle decreases. As has been discussed before, the fourth leaching cycle also corresponds to the residual reaction period, where the majority of the rare earth cations have been exchanged. With the depletion of RE cations, the seepage rate in the ore sample increases. Hence, the leaching time difference between these two tests decreases. As the concentration of the NH$_4$Cl leaching solution is quite low, the viscosity difference between the NH$_4$Cl solution and DI water can be neglected. The slower seepage rate in the 2% NH$_4$Cl leaching test is possibly induced by the porous structure evolution of the sample during the ion exchange process.

## 3.3. Inversion diagrams of pore structures during leaching tests

Figure 8*a,b* shows the inversion diagrams of porous structure evolutions of sample 1 and sample 2 during the DI water leaching test and 2% NH$_4$Cl leaching test, respectively. These diagrams were obtained via NMR imaging technology. As shown in figure 8*a*, the first inversion diagram shows a bright white colour in the most area, which indicates that the porous structure in the sample is filled with water after the saturation period (2.7 h). As the leaching process proceeded (2.7–5.1–6.9 h), inversion diagrams exhibit little change. This phenomenon indicates, with the ion exchange process, the porous structure of the sample will not be changed by the leaching process. For sample 2, the first inversion diagram after saturation period is identical to the inversion images of sample 1 (figure 8*a,b*). As the leaching process proceeded, an abnormality appears in the inversion diagram, highlighted in a red rectangular box (shown in the second image in figure 8*b*). In the NMR inversion diagram, the black colour means the void area in the sample. As discussed in §3.1 of this manuscript, the main ion exchange process takes place in this period. Therefore, the ion exchange process leads to the flooded pores in the sample by microparticles, which further lead to a decrease in the seepage rate. Indicated by the black band pattern in the second image of figure 8*b*, the main area of ion exchange was still in the upper part of the sample. In the next leaching cycle (6.9 h), the large black area in the inversion diagram has disappeared. The majority part of this inversion diagram turns back to white again. According to the discussion in §3.1, with the depletion of RE cations, the permeability of the RE ore returns to the original state. The main time period of ion exchange is between the third leaching cycle and the fourth leaching cycle.

## 3.4. SEM test of sample during leaching tests

As discussed in §3.3, ion exchange of the 2% NH$_4$Cl leaching test occurred in the third leaching cycle (3.1–5.1 h). The chemical character and surface morphology of the black strip area in the pore structure inversion image were studied using an SEM. Additional SEM tests were also performed at the same locations of the samples with two and four NH$_4$Cl leaching cycles as well as the samples leached by DI water. The test results are shown in figure 9, in which figure 9*a–c* shows electron microscopy images of the DI water leached samples with a leaching time of 2.7 h, 4.4 h, 6.4 h, respectively; figure 9*d–f* shows the electron microscopy images of the 2% NH$_4$Cl solution-leached samples with a leaching time of 3.1 h, 5.1 h and 6.9 h, respectively. The test result is magnified 20 000 times, and the solid fine particles with less

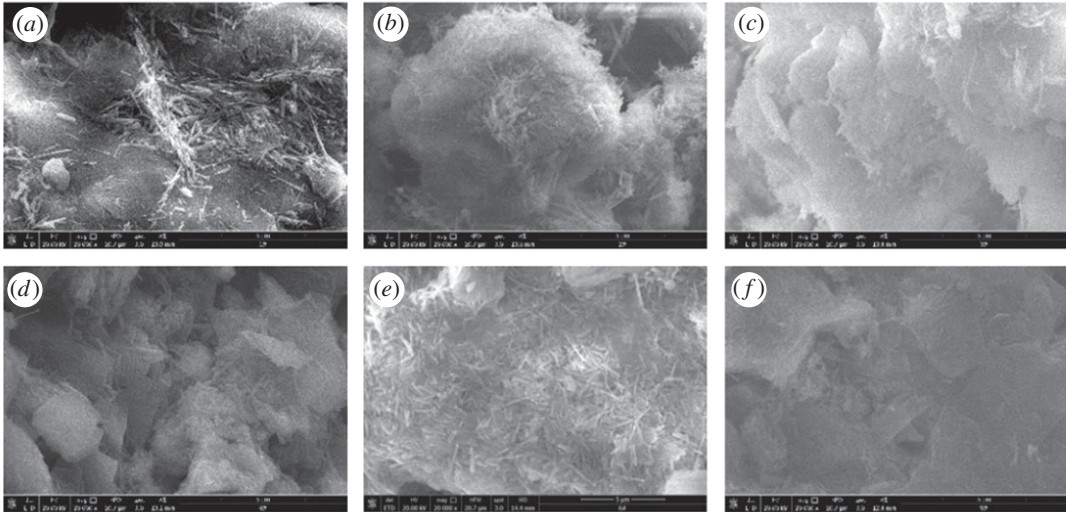

**Figure 9.** SEM diagrams of the samples.

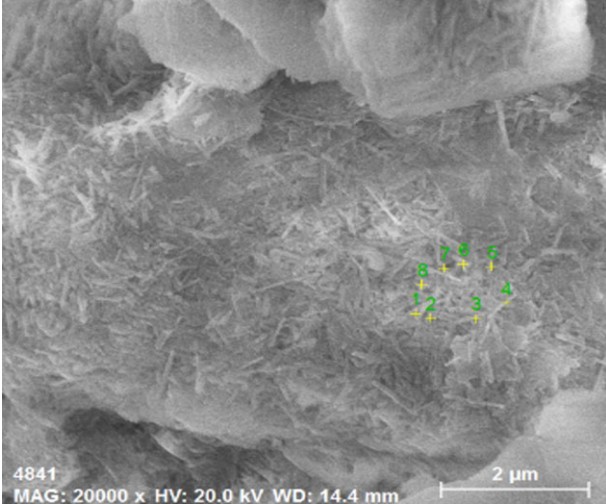

**Figure 10.** The EDS sampling positions of the soil sample.

surface distribution on the surface of the ore body leached by DI water, and the number of solid fine particles does not increase with the increase of the leaching time, which remains basically unchanged. Figure 9d shows the image after DI water saturation that is similar to figure 9a. A small amount of solid fine particles are distributed on the surface of the sample, and the particles and pores inside the sample can be clearly seen. When leaching with 2% $NH_4Cl$ solution (leaching time from 3.1 to 5.1 h), figure 9e shows that a large number of fine rod-shaped particles appear on the mineral surface of the sample, shielding the mineral surface and pores. Continuous leaching, when the ion exchange is over (6.9 h), the original large number of fine rod-shaped particles disappeared, and the mineral surface and pores reappeared. This result is consistent with the inversion image of the sample leaching process. In sum, the adsorption of fine particles on the mineral surface inside the sample is closely related to ion exchange.

On the fine particles appearing in figure 9e, eight different rod-shaped bodies were randomly selected for energy spectrum analysis and detection. The arrangement of the measuring points is shown in figure 10, and the detection results are shown in table 3. The test results show that the main components of the aggregated solid fine particles are N, O, F, Al and Si. Since the sample is fixed with epoxy resin, N and F are partial constituent elements of the epoxy resin and, therefore, are not considered. In this experiment, the total recovery of RE is 91.757%. Almost all of the RE ions in ion form have been leached. The remaining RE elements exist in the form of water-soluble phase, mineral phase and colloidal sedimentary phase. These RE elements cannot be extracted by leaching. Besides, the content of these RE elements is too low to be detected by a spectrometer. The ion-adsorbed RE ore is mainly composed of

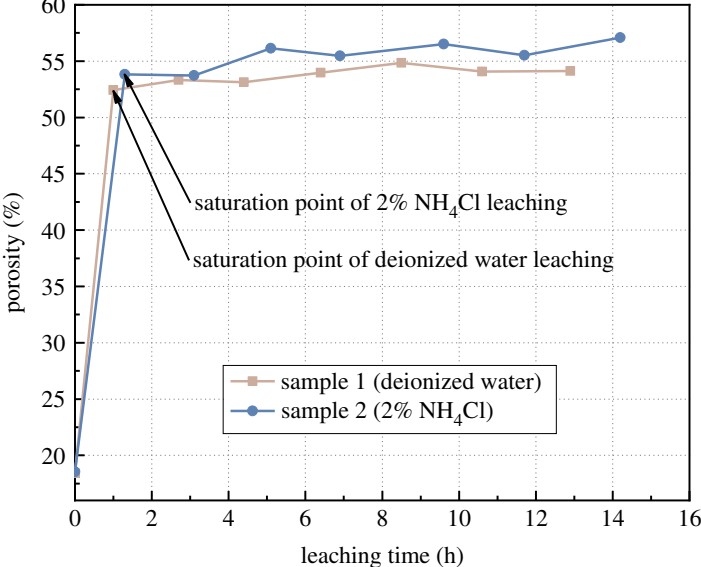

**Figure 11.** Porosity variation curve of RE specimen.

**Table 3.** Elemental mass % distribution at each measuring point.

| spectrum | N | O | F | Al | Si |
|---|---|---|---|---|---|
| 1 | 13.80 | 69.01 | 1.23 | 9.07 | 6.88 |
| 2 | 14.04 | 69.34 | 0.00 | 9.54 | 7.09 |
| 3 | 13.05 | 69.85 | 1.04 | 9.32 | 6.74 |
| 4 | 13.04 | 69.81 | 2.43 | 8.32 | 6.40 |
| 5 | 13.60 | 69.48 | 2.90 | 7.88 | 6.14 |
| 6 | 12.77 | 69.16 | 1.75 | 9.31 | 7.00 |
| 7 | 14.05 | 69.57 | 0.00 | 9.43 | 6.95 |
| 8 | 13.57 | 70.06 | 1.07 | 8.83 | 6.47 |
| mean value | 13.49 | 69.41 | 1.43 | 9.96 | 6.71 |
| sigma | 0.48 | 0.60 | 1.15 | 0.59 | 0.34 |
| sigma mean | 0.17 | 0.21 | 0.41 | 0.21 | 0.12 |

clay minerals, quartz sand, etc. The clay mineral content is about 40–70%. The clay minerals are mainly halloysite, illite, kaolinite and a very small amount of montmorillonite. The main elements that compose these ores are O, Si and Al [2,3]. As shown in the EDS results, these fine particles have the same chemical composition as the ore body, indicating these fine particles are mainly clay colloidal particles, not new materials.

## 3.5. Dynamic evolution of pore structures during leaching tests

### 3.5.1. Analysis of dynamic evolution law of porosity of rare earth ore

The porosity data of samples in two different leaching tests were measured after every leaching test and plotted in figure 11. Before the experiment, these two samples showed identical porosity. According to the data, the first two cycles are the saturation period. With the seepage of the DI water, the porosity increases dramatically. In only one cycle, the porosity of samples in both tests reached the maximum, which indicated they are all in the saturation state. During the following DI water leaching test and 2% $NH_4Cl$ leaching test, the porosity of these two samples fluctuates around the maximum value with a minor change between each cycle. This result suggests that the ion exchange process has no

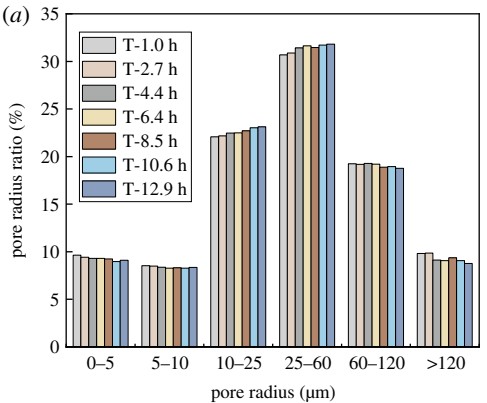 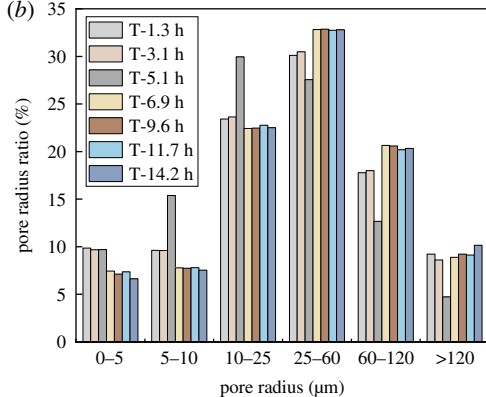

**Figure 12.** Pore radius distribution during the leaching process: (*a*) DI water leaching and (*b*) 2%NH$_4$Cl leaching.

**Table 4.** Remoulding the ratio of pore radius of RE specimen.

| pore radius (μm) | 0–5 | 5–10 | 10–25 | 25–60 | 60–120 | >120 |
|---|---|---|---|---|---|---|
| sample 1 | 35.090 | 27.703 | 25.187 | 8.747 | 1.973 | 1.301 |
| sample 2 | 34.820 | 28.280 | 25.878 | 8.491 | 1.744 | 1.086 |

influence on the porosity of samples. This result is not able to explain the reduction of the seepage rate. Therefore, the dynamic evolution of porous structures of ore samples during the leaching process needs to be investigated.

### 3.5.2. Analysis of dynamic evolution of pore structures of rare earth ore

In this part of the study, porous structure distributions of ore samples during DI water leaching test and 2% NH$_4$Cl leaching test are measured by the NM-60-type NMR microstructure analyser. Based on the size of the half radius of the pore structure, pores are divided into six different categories: 0–5 μm is a minimal pore, 5–10 μm is a small pore, 10–25 μm is a medium pore, 25–60 μm is a medium–large pore, 60–120 μm is a large pore and greater than 120 μm is a mega pore. As shown in table 4, sample 1 and sample 2 have almost the same percentage of pore structures in these six categories. Figure 12 shows the distribution of different porous structures in the saturation period and the leaching period. Figure 12*a* is the evolution of the pore structure of the of the DI water leaching process. The result shows that the porous distribution in sample 1 remains intact during the experiment process. Figure 12*b* is the pore size distribution diagram of sample 2 in 2% NH$_4$Cl leaching test. The first two cycles belong to the saturation period, where the pore size distribution remains the same as sample 1. As discussed in §3.1, the third cycle of 2% NH$_4$Cl leaching test is the period where the majority of the RE cations are replaced (5.1 h). With the NMR inspection, the data reveal that the number of pores with size under 25 μm (micro-pores, small pores and medium pores) increases dramatically and the number of pores with size above 25 μm (large pores and mega pores) decreases significantly. The fourth cycle is the residual period of the ion exchange process. This period shows an opposite trend with the incline of the number of pores with size under 25 μm (micro-pores, small pores and medium pores) and decline of the number of pores with size above 25 μm (large pores and mega pores). The pore size distribution maintains constant in the following leaching cycles. The ion exchange process induces the reduction in the number of large pores and mega pores and the rise in the number of small pores and medium pores. With the depletion of the RE cations, the pore structure of the sample changes back from small pore size to large pore size and the whole porous structure of the ore body back to the original state.

## 4. Discussion

The *in situ* leaching is a common method of extracting RE elements from the ion-adsorbed RE ore. The ore body acts as an aqueous medium, and its pore structure is the main pass way for the seepage of

leaching solutions. The pore structure evolution of the ore body determines the extracting rate of RE resources. The migration, deposition and release processes of fine particles in aqueous media are affected by many factors, such as temperature, pH and concentration of leaching solution [37,38]. The *in situ* leaching mining is coupled with the solution seepage and the ion exchange process [39]. In this study, we focused on the influence of the leaching process on the pore structure of the ore body. All samples were first saturated with DI water before the experiment. Hence, at the beginning of the injection, the samples were in the seepage state. On the one hand, the samples were washed by the leaching action of DI water. As a result, each sample was clean, and the influence of other factors could be ruled out. On the other hand, after the samples were saturated, the internal percolation channels of the sample were all connected and the factors that influence the change of the pore structure of the sample were the migration, deposition and release of the fine particles. The room temperature and the concentration of the leaching solutions were constant in this test. The pH difference between the leaching solution and the recovered liquid was found to be very small, indicating that the pH inside the ore body remains constant during the whole leaching process. By comparing with the control experiment, it has been proved that the ion exchange process reduces the seepage rate of the leaching solution in the ore body. Further investigation shows the ion exchange process has no influence on the porosity. However, the pore structure distribution in the ore body will be changed at different stages of the leaching process. Specifically, the ion exchange process induces the shrinkage of the pore size at the beginning and with the finishing of the ion exchange process, the pore size evolves from a medium/small pore structure to a large pore structure and the pore structure of the ore body changes back to its original state. The mechanism of the pore structure evolution is due to the deposition and releasing of microparticles on pore structures in the ore body, which is caused by the ionic strength change in the $NH_4Cl$ leaching solution during the leaching process. In the initial ion exchange process, cations with +1 charge are replaced by cations with +3 charges and the ionic strength of the leaching solution increases. The electric double layer of the clay colloidal particles inside the sample is compressed, causing the Van der Waals attraction, the electric double layer repulsion between the colloidal particles and the mineral surface to be out of balance [40,41]. The increase in the ionic strength induces the deposition of a large number of microparticles in the surface of the ore body, blocking the pore structures. The size pore structure of the ore body evolves from a large pore structure to a medium/small pore structure. Along with the finishing of ion exchange reaction, the majority of the RE cations with +3 charges have been leached from the ore body. With constant seepage of $NH_4Cl$ leaching solution, the ionic strength of the leaching solution in the pore structure of the ore body is reduced as RE cations with +3 charges are constantly replaced by cations with +1 charge. The thickness of the electric double layer of the clay colloidal particles inside the sample increases again, and the electric double layer repulsive force once again becomes the dominant force [42,43]. The microparticles absorbed around large pores are released and large pores change back to their initial state and the pore structure of the whole ore body shows a transition from a medium/ small pore structure to a large pore structure. Therefore, the ion exchange of the RE ore body leaching process induces the deposition and release of fine particles inside the ore body, resulting in the dynamic evolution of the pore structure. This phenomenon inhibits the seepage of the leaching solution in the ore body to a certain extent and slows down the collection rate of the mother solution.

# 5. Conclusion

(1) The ion exchange process hinders the seepage of the leaching solution. With the completion of the ion exchange process, the seepage rate gradually recovers back to its original level. The seepage rate difference between two leaching liquors can be determined by measuring the recovery time of the same amount of leaching liquor. The DI water leaching process does not involve the ion exchange process. With the injection of the same amount of leaching solution, the time interval for recovering the same amount of leaching solution remains constant. In the 2% $NH_4Cl$ leaching test, the collecting time of every leaching cycles is different from that in the DI water test. The time difference between these two tests raises at the beginning of the leaching cycle and then drops afterward.

(2) The ion exchange process does not influence the porosity of the ore body. During the ion exchange process, the pore structure of the ore body shows an evolution from a large pore structure to a medium/small pore structure, where the number of large pores decreases and the number of small pores and medium pores increases. With the completion of the ion exchange process, the pore structure evolves from a medium/small pore structure to a large pore structure and the pore structure

of the ore body returns to its original state. The dynamic evolution of porous structures suppresses the leaching solution seepage, resulting in the reduction of the collecting rate of mother solutions.

(3) The ion exchange between the leaching solution and the ore body induces the increase of the ionic strength of the leaching solution, which causes the adsorption of a large amount of fine clay colloidal particles on the inner surface of the ore sample. As a result, large pores in the sample are blocked and the seepage rate of the leaching solution is reduced. After the completion of ion exchange, under the seepage of leaching solution, the ionic strength of the leaching solution is reduced, which leads to the desorption of the fine clay colloidal particles, resulting in the restoration of the pore structure and the recovery of the leaching rate.

Data accessibility. Data available from the Dryad Digital Repository: https://doi.org/10.5061/dryad.05qfttdz1 [44].

Authors' contributions. L.Z. and X.W. designed the study and drafted the manuscript. L.Z., Y.Z. and G.H. carried out the laboratory work. K.H. and W.Z. participated in data analysis. All authors gave final approval for publication and agreed to be held accountable for the work performed therein.

Competing interests. We declare we have no competing interests.

Funding. This work was supported by the National Natural Science Foundation of China (grant nos. 51564012, 51874148, 51504102 and 51764014); the Science and Technology Project Funded by the Education Department of Jiangxi Province (grant no. GJJ150653); the Qingjiang Excellent Young Talents of Jiangxi University of Science and Technology, the National Key R&D Program of China (grant no. 2017YFC0804601); the Jiangxi University of Science and Technology Excellent Doctoral Thesis Training Project (grant no. YB2017001) and the Young Jinggang Scholars Award Program in Jiangxi Province.

Acknowledgements. We thank Hao Wang and Yongxin Li for their assistance with the data collection. We also thank anonymous reviewers and editors for their insightful suggestions and careful reading.

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
