## [Reviewer comments · Royal Society Open Science]

Review History

RSOS-191107.R0 (Original submission)

Review form: Reviewer 1

Is the manuscript scientifically sound in its present form?

No

Are the interpretations and conclusions justified by the results?

No

Is the language acceptable?

Yes

Do you have any ethical concerns with this paper?

No

Have you any concerns about statistical analyses in this paper?

No

Recommendation?

Major revision is needed (please make suggestions in comments)

Comments to the Author(s)

The evaluated paper deals with the results of the ion adsorbed rare earth ore leaching with water and salt. The main point of my review is the goal of the study. Authors should better articulate the goal of the research. At the moment the readers do not know why the studies were done.

Why the study of the texture of the extracted material is so important? Where can this knowledge be used? Regardless of the above, I have other comments.

1. I did not find the results of the chemical and mineralogical analysis of the samples. Authors describe the details of the samples after reaction with DW and NH₄Cl using EDS and SEM. But I do not know the chemical composition of the ore samples and above all the mineralogical composition. It is extremely hard to discuss the results without the knowledge of the mineralogical composition.

2. The authors assumed that rare earth cations are adsorbed on the exchange position. Did they do the full analysis of the RE mineral speciation in the samples? I did not find such results. RE can form another forms, can be adsorbed with Fe, Mn, Al oxides (if occur in the sample) or occur in the structure others minerals in the ore. But without of the chemical, mineralogical analysis results it is hard to say anything.

3. What is the reason for the observed textural changes? The only ion exchange reaction or also the reaction NH₄Cl with the sample in general? Authors do not discuss this matter in the manuscript.

4. Table 1. What does it mean "Proportion"? Which proportion?

5. Page 5, Line 27. "...sampling E mL". What does it mean "E"?

6. Table 3. O, Al, Si are the only elements in the samples? Are you sure? Please present the EDS spectrum.

7. The results from the figure 12 which should be the quintessence of the obtained results. I did not find any spectacular results. A lot of results discussed in the manuscript but without significant conclusions.

Review form: Reviewer 2

Is the manuscript scientifically sound in its present form?

No

Are the interpretations and conclusions justified by the results?

Yes

Is the language acceptable?

No

Do you have any ethical concerns with this paper?

No

Have you any concerns about statistical analyses in this paper?

No

Recommendation?

Major revision is needed (please make suggestions in comments)

Comments to the Author(s)

In this manuscript authors have studied the effect of pore structures in the ion exchange on rare earth ores. But there are several areas where thorough revisions are necessary before this manuscript can be considered for publication. See the comments below:

Introduction is too brief. Importance of ion exchange should be discussed in-depth. See and include this work: J. Mater. Chem. 2009, 19, 1901-1907.

Authors have state the dimension of micropores as 'micropores (0~5 μ m)' But this is against conventional nomenclature for microporous and mesoporous materials. Terminology should be clear.

There are several grammatical errors, these should be corrected.

The quality of SEM images is not good. HR TEM analysis may provide useful information on the porosity in these materials.

Decision letter (RSOS-191107.R0)

02-Sep-2019

Dear Professor Wang:

Title: Dynamic pore structure evolution of the ion adsorbed rare earth ore during the ion exchange process

Manuscript ID: RSOS-191107

The editor assigned to your manuscript has now received comments from reviewers. We would like you to revise your paper in accordance with the referee and Subject Editor suggestions which can be found below (not including confidential reports to the Editor). Please note this decision does not guarantee eventual acceptance.

Please submit your revised paper before 25-Sep-2019. Please note that the revision deadline will expire at 00.00am on this date. If we do not hear from you within this time then it will be assumed that the paper has been withdrawn. In exceptional circumstances, extensions may be possible if agreed with the Editorial Office in advance. We do not allow multiple rounds of revision so we urge you to make every effort to fully address all of the comments at this stage. If deemed necessary by the Editors, your manuscript will be sent back to one or more of the original reviewers for assessment. If the original reviewers are not available we may invite new reviewers.

RSC Associate Editor:
Comments to the Author:
(There are no comments.)

RSC Subject Editor:
Comments to the Author:
(There are no comments.)

Reviewers' Comments to Author:
Reviewer: 1

Comments to the Author(s)

The evaluated paper deals with the results of the ion adsorbed rare earth ore leaching with water and salt. The main point of my review is the goal of the study. Authors should better articulate the goal of the research. At the moment the readers do not know why the studies were done. Why the study of the texture of the extracted material is so important? Where can this knowledge be used? Regardless of the above, I have other comments.

1. I did not find the results of the chemical and mineralogical analysis of the samples. Authors describe the details of the samples after reaction with DW and NH₄Cl using EDS and SEM. But I do not know the chemical composition of the ore samples and above all the mineralogical composition. It is extremely hard to discuss the results without the knowledge of the mineralogical composition.
2. The authors assumed that rare earth cations are adsorbed on the exchange position. Did they do the full analysis of the RE mineral speciation in the samples? I did not find such results. RE

can form another forms, can be adsorbed with Fe, Mn, Al oxides (if occur in the sample) or occur in the structure others minerals in the ore. But without of the chemical, mineralogical analysis results it is hard to say anything.

3. What is the reason for the observed textural changes? The only ion exchange reaction or also the reaction NH_4Cl with the sample in general? Authors do not discuss this matter in the manuscript.

4. Table 1. What does it mean "Proportion"? Which proportion?

5. Page 5, Line 27. "...sampling E mL". What does it mean "E"?

6. Table 3. O, Al, Si are the only elements in the samples? Are you sure? Please present the EDS spectrum.

7. The results from the figure 12 which should be the quintessence of the obtained results. I did not find any spectacular results. A lot of results discussed in the manuscript but without significant conclusions.

Reviewer: 2

Comments to the Author(s)

In this manuscript authors have studied the effect of pore structures in the ion exchange on rare earth ores. But there are several areas where thorough revisions are necessary before this manuscript can be considered for publication. See the comments below:

Introduction is too brief. Importance of ion exchange should be discussed in-depth. See and include this work: J. Mater. Chem. 2009, 19, 1901-1907.

Authors have state the dimension of micropores as 'micropores ($0\sim 5\ \mu\text{m}$)' But this is against conventional nomenclature for microporous and mesoporous materials. Terminology should be clear.

There are several grammatical errors, these should be corrected.

The quality of SEM images is not good. HR TEM analysis may provide useful information on the porosity in these materials.

Author's Response to Decision Letter for (RSOS-191107.R0)

See Appendix A.

RSOS-191107.R1 (Revision)

Review form: Reviewer 1

Is the manuscript scientifically sound in its present form?

Yes

Are the interpretations and conclusions justified by the results?

Yes

Is the language acceptable?

Yes

Do you have any ethical concerns with this paper?

No

Have you any concerns about statistical analyses in this paper?

No

Recommendation?

Accept as is

Comments to the Author(s)

The authors improved the manuscript in a significant way. In my opinion, it can be published in the current form.

Review form: Reviewer 2

Is the manuscript scientifically sound in its present form?

Yes

Are the interpretations and conclusions justified by the results?

Yes

Is the language acceptable?

Yes

Do you have any ethical concerns with this paper?

No

Have you any concerns about statistical analyses in this paper?

No

Recommendation?

Accept as is

Comments to the Author(s)

Authors have addressed the referee comments properly and the manuscript has been improved. Now this revised version of the manuscript may be accepted in its current form.

Decision letter (RSOS-191107.R1)

04-Oct-2019

Dear Professor Wang:

Title: Dynamic pore structure evolution of the ion adsorbed rare earth ore during the ion exchange process

Manuscript ID: RSOS-191107.R1

It is a pleasure to accept your manuscript in its current form for publication in Royal Society Open Science. The chemistry content of Royal Society Open Science is published in collaboration with the Royal Society of Chemistry.

RSC Associate Editor:
Comments to the Author:
(There are no comments.)

RSC Subject Editor:
Comments to the Author:
(There are no comments.)

Reviewer(s)' Comments to Author:
Reviewer: 2

Comments to the Author(s)
Authors have addressed the referee comments properly and the manuscript has been improved. Now this revised version of the manuscript may be accepted in its current form.

Reviewer: 1

Comments to the Author(s)
The authors improved the manuscript in a significant way. In my opinion, it can be published in the current form.

Appendix A

List of Responses

Dear Editors and Reviewers:

Thank you for your letter and for the reviewers' comments concerning our manuscript entitled "Dynamic pore structure evolution of the ion adsorbed rare earth ore during the ion exchange process" (ID: RSOS-191107). Those comments are all valuable and very helpful for revising and improving our paper, as well as the important guiding significance to our researches. We have studied comments carefully and have made correction which we hope meet with approval. Revised portion are marked in red in the paper. The main corrections in the paper and the responds to the reviewer's comments are as flowing:

Responds to the reviewers' comments:

Reviewer #1:

1. At the moment the readers do not know why the studies were done. Why the study of the texture of the extracted material is so important? Where can this knowledge be used?

Response: Considering your suggestion, we have added related content in the introduction section in paragraph 2 on page 2, and the revised part has been marked in red in the manuscript.

By comparing and analyzing the changes of internal pore structure of rare earth ore samples during the leaching process with deionized water and 2% NH₄Cl solution, we have revealed the effect of ion exchange on the pore structure of ore bodies. In the in-situ leaching process, the change of the pore structure of the ore body is closely related to the seepage of the rare earth mother liquor. On the one hand, the percolation rate of the rare earth mother liquor is related to the extraction rate of the rare earth element. On the other hand, the stability of the rare earth ore body under the seepage is extremely poor, which affects the stability of the hillsides of rare earth mines. Therefore, our research provides useful guidelines for efficient development and safe production of rare earth mines

2. I did not find the results of the chemical and mineralogical analysis of the samples. Authors describe the details of the samples after reaction with DW and NH₄Cl using EDS and SEM. But I do not know the chemical composition of the ore samples and above all the mineralogical composition. It is extremely hard to discuss the results without the knowledge of the mineralogical composition.

Response: It is very important to fully understand the mineral composition of the rare earth ore, and further distinguish which component will chemically react with the leaching solution, and how they react. If this problem is thoroughly studied, it will be very important for the extraction of rare earth elements. However, in the content of this research, we focused on the extraction of rare earth elements, and the specific chemical reactions involved in the leaching process were not deeply investigated. The chemical reaction of the leaching process is shown in Equ.1 in manuscript, which is also supported by related research papers.

Before the start of the experiment, we tested the content of each element in the rare earth ore, the total amount of rare earth oxides, and the rare earth oxide distribution. The content of each element in rare earth ore powder sample was analyzed by the Axios max X-ray fluorescence spectrometer and the test results are shown in Table 1. The total amount and distribution of rare earth oxides were measured by Agilent 8800 inductively coupled plasma mass spectrometer and the test results are shown in Table 2. As shown in Table 1, the main elements in rare earth ore samples are O, Al, Si, and the fractions of other elements are basically trace amounts, and the only rare earth element that can be detected is Y. However, based on Table 2, other rare earth elements are also present. In this research field, all rare earth elements are denoted as RE³⁺, and the equation of the chemical reaction during the leaching process is shown in Eq.1. Other reactions must exist, but they are negligible as they are not the focus of this research. For this reason, we did not report the mineral composition of rare earth ore in the paper.

Considering your suggestion, we have revised the sections 2.1 and 3.4 of this paper in paragraph 3 on page 2 and paragraph 2 on page 8, and the revised part has been marked in red in the manuscript.

The supplementary data is not convenient to put in the body of the paper, we have submitted this data package to Dryad as the supporting information. The temporary review link: https://datadryad.org/stash/share/XdWWim9OIX5CWVRjnhn_HpIHjhy0Lq29gpexZQh_Ank.

Table 1. Contents of elements in rare earth ore powder samples

Analyte	Calibration status	Compound formula	Concentration	Unit	Calculation method	Status
O	Calibrated	O	39.985	%	Calculate	BgC;DC;
F	Calibrated	F	0.164	%	Calculate	BgC;DC;
Na	Calibrated	Na	0.136	%	Calculate	BgC;DC;

Al	Calibrated	Al	12.435	%	Calculate	BgC;DC;
Si	Calibrated	Si	26.388	%	Calculate	BgC;DC;
P	Calibrated	P	0.008	%	Calculate	BgC;DC;
S	Calibrated	S	0.013	%	Calculate	BgC;DC;
Cl	Calibrated	Cl	0.012	%	Calculate	BgC;DC;
K	Calibrated	K	4.800	%	Calculate	BgC;DC;
Ca	Calibrated	Ca	0.036	%	Calculate	BgC;DC;
Ti	Calibrated	Ti	0.015	%	Calculate	BgC;DC;
Mn	Calibrated	Mn	0.069	%	Calculate	BgC;DC;
Fe	Calibrated	Fe	1.005	%	Calculate	BgC;DC;LoR;
Zn	Calibrated	Zn	0.013	%	Calculate	BgC;DC;
Ga	Calibrated	Ga	0.005	%	Calculate	BgC;DC;
As	Calibrated	As	0.005	%	Calculate	BgC;DC;
Rb	Calibrated	Rb	0.119	%	Calculate	BgC;DC;LoR;
Y	Calibrated	Y	0.020	%	Calculate	BgC;DC;
Zr	Calibrated	Zr	0.007	%	Calculate	BgC;DC;
W	Calibrated	W	0.010	%	Calculate	BgC;DC;
Pb	Calibrated	Pb	0.021	%	Calculate	BgC;DC;
Th	Calibrated	Th	0.003	%	Calculate	BgC;DC;IC;
	Sum		85.3	%	-	-

Table 2 Total amount and weight percentage of rare earth elements in rare earth ore powder samples (%)

Analyte	Y ₂ O ₃	La ₂ O ₃	CeO ₂	Pr ₆ O ₁₁	Nd ₂ O ₃	Sm ₂ O ₃	Eu ₂ O ₃
Sample 1	0.0256	0.00492	0.0132	0.00171	0.00717	0.00333	0.000076
Sample 2	0.0244	0.00454	0.0077	0.0016	0.00661	0.00308	0.00007
Sample 3	0.0253	0.00478	0.00905	0.00166	0.00687	0.0032	0.000074
Average	0.0251	0.00475	0.00998	0.00166	0.00688	0.0032	0.000073

Continue the table:

Gd ₂ O ₃	Tb ₄ O ₇	Dy ₂ O ₃	Ho ₂ O ₃	Er ₂ O ₃	Tm ₂ O ₃	Yb ₂ O ₃	Lu ₂ O ₃	REO
0.00386	0.00071	0.00432	0.00087	0.00257	0.00041	0.00294	0.00044	0.07213
0.00355	0.00065	0.00406	0.00083	0.00246	0.0004	0.00284	0.00042	0.06321
0.00369	0.00068	0.0042	0.00085	0.00251	0.00041	0.00288	0.00042	0.06657
0.0037	0.00068	0.00419	0.00085	0.00251	0.00041	0.00289	0.00043	0.0673

3. The authors assumed that rare earth cations are adsorbed on the exchange position. Did they do the full analysis of the RE mineral speciation in the samples? I did not find such results. RE can form another forms, can be adsorbed with Fe, Mn, Al oxides (if occur in the sample) or occur in the structure others minerals in the ore. But without of the chemical, mineralogical analysis results it is hard to say anything.

Response: The problem you mentioned does exist that the RE can form another forms as it can be adsorbed by Fe, Mn, Al oxides. However, the rare earth leaching rate of this test reached 91.757%, and most of the rare earth elements have been rinsed out. The grade of the ion-adsorbed rare earth ore is about 0.05%~0.3%. Among them, 75%~95% of the rare earth elements are in the form of ions, and the remaining rare earth elements are in the form of water-soluble phase, mineral phase and colloidal sedimentary phase. In this study, we have used 2% ammonium chloride solution to leach, and the rare earth elements that can be extracted are all ionic phases. Ionic phase refers to rare earth ions that are adsorbed on the surface of clay minerals in the form of hydrated ionic or hydroxy hydrated ionic ions. These rare earth ions are chemically stable and do not hydrolyze in natural water. But when they encounter more reactive ions, such as NH₄⁺, Mg²⁺, etc, they will be displaced from the surface of clay minerals. This comment has something in common with the 3rd comment, but what you mentioned is not the focus of this research. The focus of this study is on the influence of chemical substitution on the pore structure of rare earth samples.

We have added related content in paragraph 2 on page 8, and the revised part has been marked in red in the manuscript.

4. What is the reason for the observed textural changes? The only ion exchange reaction or also the

reaction NH_4Cl with the sample in general? Authors do not discuss this matter in the manuscript.

Response: In the discussion section of the paper we have added relevant content.

All samples were first saturated with deionized water before the experiment. Hence, that the beginning of the injection, the samples were in the seepage state. On the one hand, the samples were washed by the leaching action of deionized water. As a result, each sample was clean, and the influence of other factors could be ruled out. On the other hand, after the samples were saturated, the internal percolation channels of the sample were all connected, and the factors that affecting of the pore structure of the sample were the migration, deposition, and release of fine particles. The migration, deposition and release processes of fine particles in aqueous media are affected by many factors, such as temperature, pH, concentration of leaching solution. In this study, we focused on the influence of the leaching process on the pore structure of the ore body. The room temperature and the concentration of the leaching solutions were constant in this test. The pH value difference between the leaching solution and the recovered liquid was found to be very small, indicating that the pH inside the ore body remains constant during the whole leaching process. With these well controlled factors, we discussed the dynamic evolution of the pore structure of the sample and its mechanism.

Considering your suggestion, we have added related content in paragraph 2 on page 10 and paragraph 2 on page 8, and the revised part has been marked in red in the manuscript.

5. Table 1. What does it mean “Proportion”? Which proportion?

Response: We are very sorry for our incorrect writing. After discussion, “proportion” means “a part, share, or number considered in comparative relation to a whole.” What we want to express in the article is a physical parameter of rare earth ore. The correct term is specific density. We have made changes in the manuscript.

6. Page 5, Line 27. “...sampling E mL”. What does it mean “E”?

Response: We are very sorry about this. E is the volume of the sampled mother solution, in order to avoid misunderstanding, it has been modified to V_3 .

7. Table 3. O, Al, Si are the only elements in the samples? Are you sure? Please present the EDS spectrum.

Response: We studied the fine rod-shaped particles observed in the SEM. To ensure the accuracy of the test, we took several samples for testing, and the results were basically the same. The testing locations and EDS spectrum of sample 1 are shown in Figure 1 and the test results are shown in Table 3. The testing locations and EDS spectrum of sample 2 are shown in Figure 2 and the test results are shown in Table 4. The testing locations and EDS spectrum of sample 3 are shown in Figure 3 and the test results are shown in Table 5. After discussion within the research team, we selected the most representative one for analysis. On the one hand, according to Table 1, except for the high content of O, Al and Si, the other elements are basically trace amounts, and the trace elements may not be detected by the EDS. On the other hand, it is very common to use a scanning electron microscope to observe a sample, and the selected sample portion has a large randomness, and other elements are likely existed. Our expression in the text is that the main constituent elements of solid fine particles are O, Si, Al, and may also contain other trace elements. It does not mean that O, Si and Al are the only elements. Previous studies have shown that, the mineral composition of ion-adsorbed rare earth ore is mainly composed of clay minerals, quartz sand, etc. The clay mineral content is about 40% to 70%, and it is mainly composed of halloysite, illite, kaolinite and a very small amount of montmorillonite. Besides, the main constituent elements of these ores are O, Si and Al. This result indicates that the main constituent elements of solid fine particles are O, Si, and Al, which are the main constituent elements of clay minerals. Hence, we concluded that these fine particles are not new materials, but mainly clay colloidal particles.

Considering your suggestion, we have added related content in paragraph 2 on page 7, and the revised part has been marked in red in the manuscript.

We have submitted this data package to Dryad as the supporting information. The temporary review link: https://datadryad.org/stash/share/XdWWim9OIX5CWVRjnhn_HpIHjhy0Lq29gpexZQh_Ank.

(a) The EDS sampling positions of sample 1.

(b) the EDS spectrum of sample 1

Figure1. The EDS sampling positions and the EDS spectrum of sample 1

Table 3. Elemental mass fraction at each measuring point of sample 1

Spectrum	N	O	F	Al	Si
1	14.14	68.23	1.02	9.49	7.13
2	13.86	67.64	1.56	9.76	7.17
3	13.70	65.53	3.91	9.76	7.11
4	14.28	67.95	0.59	9.70	7.48
5	13.68	66.78	3.16	9.32	7.06
6	14.49	68.61	1.71	8.58	6.62
Mean value	14.02	67.46	1.99	9.43	7.09
Sigma	0.33	1.13	1.28	0.46	0.28
Sigma mean	0.13	0.46	0.52	0.19	0.11

(a) The EDS sampling positions of sample 2.

(b) the EDS spectrum of sample 2

Figure2. The EDS sampling positions and the EDS spectrum of sample 2

Table 4. Elemental mass fraction at each measuring point of sample 2

Spectrum	N	O	F	Al	Si
1	14.07	69.32	0.00	9.50	7.12
2	13.50	66.01	4.43	9.36	6.70
3	13.43	68.99	1.45	9.27	6.86
4	14.04	69.85	1.40	8.26	6.45
5	14.02	70.07	0.69	8.63	6.60
6	13.99	68.99	0.00	9.69	7.33
7	14.19	67.72	2.23	8.81	7.06
Mean value	13.89	68.71	1.46	9.07	6.87
Sigma	0.30	1.41	1.54	0.52	0.32
Sigma mean	0.11	0.53	0.58	0.20	0.12

(a) The EDS sampling positions of sample 3.

(b) the EDS spectrum of sample 3

Figure3. The EDS sampling positions and the EDS spectrum of sample 3

Table 5. Elemental mass fraction at each measuring point of sample 3

Spectrum	N	O	F	Al	Si
1	13.80	69.01	1.23	9.07	6.88
2	14.04	69.34	0.00	9.54	7.09
3	13.05	69.85	1.04	9.32	6.74
4	13.04	69.81	2.43	8.32	6.40
5	13.60	69.48	2.90	7.88	6.14
6	12.77	69.16	1.75	9.31	7.00
7	14.05	69.57	0.00	9.43	6.95
8	13.57	70.06	1.07	8.83	6.47
Mean value	13.49	69.41	1.43	9.96	6.71
Sigma	0.48	0.60	1.15	0.59	0.34
Sigma mean	0.17	0.21	0.41	0.21	0.12

8. The results from the figure 12 which should be the quintessence of the obtained results. I did not find any spectacular results. A lot of results discussed in the manuscript but without significant conclusions.

Response: The content in Section 3.4.2 is indeed the quintessence of our research. The whole paper is built around the results. Section 3.1 describes that the ion exchange process mainly took place in the third cycle and the fourth cycle, where the third cycle is the main reaction period and the fourth cycle is the residual reaction period. And it is confirmed that in the 2% NH_4Cl solution leaching process, the effective leaching time ranges from 3.1 h to 6.9 h. In Section 3.2, the factors that cause longer effective leaching time in the third and fourth cycles of the 2% NH_4Cl leaching compared with that of the deionized water leaching are discussed. It is inferred that the slower seepage rate in the 2% NH_4Cl leaching test is possibly induced by the porous structure evolution of the sample during the ion exchange process. The inversion diagrams of samples were obtained via NMR imaging technology. During the leaching process, an abnormality appeared in the inversion diagram. As been discussed in section 3.1 of this manuscript, the majority of the ion exchange process takes place in this period. Therefore, the ion exchange process leads to the blockage of pores in the sample by microparticles, which further lead to a decrease in the seepage rate. In the next leaching cycle, the large black area in the inversion diagram had disappeared and the permeability of the rare earth ore returned to the original state. Then the chemical character and surface morphology of the black strip area in the pore structure inversion image were studied using an SEM. The deionized water leached sample has a small amount of solid fine particles adsorbed on the surface. The number of the solid fine particles did not increase with the increase of leaching time. However, when leached with the 2% NH_4Cl solution, many fine rod-shaped particles appeared on the surface of the sample, shielding the mineral surface and pores. This proves the presence of solid particles mentioned Section 3.1. Different rod-shaped particles were randomly selected for energy spectrum analysis. The main constituent elements of solid fine particles are O, Si, and Al, indicating that these fine particles are not new materials, mainly clay colloidal particles. However, we observed a dynamic evolution of the porosity of the sample during the leaching process. It was found that the ion exchange does not change the overall porosity of the rare earth sample. Therefore, the pore structure of the rare earth sample during the leaching process must be studied. In Section 3.4.2, the following trend was summarized: At the beginning of the ion exchange process, the ion exchange process induces the reduction in the number of large pores and mega pores and the rise in the number of small pores and media pores; with the depletion of the rare earth cations, the pore structure of the sample changes back from small pore size to large pore size and the whole porous structure of the ore body back to the original state. Besides, in the discussion part, the mechanism of dynamic evolution of pore structure of rare earth ore bodies during leaching was analyzed in detail. Our research is mainly focused on the process of dynamic change of rare earth ore body structure in leaching and the mechanism of this dynamic change. Additionally, we also fully analyzed the chemical composition of the rare earth ore, the chemical reactions that involved in the leaching process, the chemical reaction equations, and the products of chemical reactions. As these results are outside the scope of this study, they are not discussed in this manuscript.

Considering your suggestion, we have added related content in paragraph 1 on page 10 and further refined the conclusions in paragraph 2 and 3 on page 11, and the revised part has been marked in red in the manuscript.

Once again, thank you very much for your comments and suggestions.

Reviewer #2:

1. Introduction is too brief. Importance of ion exchange should be discussed in-depth. See and include this work: J. Mater. Chem. 2009, 19, 1901-1907.

Response: Referring to your suggestion, we further discuss the importance of ion exchange in the introduction section and the proposed reference has been cited in this manuscript. And we have made revisions in paragraph 3 on page 1 and paragraph 1 on page 2, and the revised part has been marked in red in the manuscript.

2. Authors have state the dimension of micropores as “micropores (0~5 μm)” But this is against conventional nomenclature for microporous and mesoporous materials. Terminology should be clear.

Response: In this paper, we study the dynamic evolution of the pore structure of rare earth ore bodies during ion exchange and the reasons for this change. For the sake of analysis and explanation, we classify the pores according to different pore sizes. Based on the size of the half radius of the pore structure, pore structures are divided into six different categories: 0~5 μm is minimal pore, 5~10 μm is small pore, 10~25 μm is medium pore, 25~60 μm is medium-large pore, 60 μm ~120 μm is large pore, and >120 μm is mega pore.

We have read the relevant papers and made revisions in paragraph 1 on page 1 and paragraph 2 on page 9, and the revised part has been marked in red in the manuscript.

3. There are several grammatical errors, these should be corrected.

Response: We are very sorry for our negligence of these mistakes. Through careful examination, we have modified the grammatical errors in the paper.

a. “as the end of the ion exchange process, the evolution of porous structure shows an opposite trend.” have been corrected as “Along with the completion of the ion exchange process, the evolution of porous structure shows an opposite trend.”

b. “ E is the volume is the sampled mother solution, usually 5.00 mL” have been corrected as “ V_3 is the volume of the sampled mother solution, usually 5.00 mL”.

4. The quality of SEM images is not good. HR TEM analysis may provide useful information on the porosity in these materials.

Response: The surface morphology analysis of rare earth ore was carried out with a MLA650F field emission electron microscope and energy spectrometer, the problem you mentioned does exist. The image quality is limited by the performance of the SEM.

Special thanks to you for your good comments.